

# Volatile organic compounds from entomopathogenic and nematophagous fungi repel banana weevil (*Cosmopolites sordidus*) under banana field conditions

Ana Lozano-Soria[1], Ana Piedra-Buena Diaz[2], Federico Lopez-Moya[1], Miguel Valverde-Urrea[1], Jose J. Zubcoff[1], Jose Emilio Martinez-Perez[3], Javier Lopez-Cepero[4] and Luis V. Lopez-Llorca[1]

[1] Marine Sciences and Applied Biology, University of Alicante, Alicante, Spain
[2] Entomology Area, Plant Protection Department, Canarian Institute for Agronomy Research, Valle de Guerra, San Cristóbal de La Laguna, Tenerife, Canary Islands, Spain
[3] Natural Resources Cartography Unit, Department of Ecology, University of Alicante, Alicante, Spain
[4] Technical Department of Coplaca, Coplaca Cooperative, Tenerife, Canary Islands, Spain

Corresponding author
Federico Lopez-Moya,
federico.lopez@ua.es

## ABSTRACT

Fungal volatile organic compounds (VOCs) with biological activity produced by entomopathogenic fungi (*Beauveria bassiana* and *Metarhizium robertsii*) isolated from banana fields (Canary Islands) and the nematophagous fungus *Pochonia chlamydosporia*, used in biocontrol of root-knot nematodes, repel the banana weevil (BW), *Cosmopolites sordidus* (Germar, 1824) under laboratory conditions. BW is the main pest of banana (*Musa* spp. (Linnaeus, 1753)). Its cryptic behavior makes it difficult to manage. Repellent VOCs alter BW behavior and can thus be used in sustainable pest management strategies. We evaluated fungal VOCs styrene (C1), benzothiazole (C2), 1,3-dimethoxybenzene (C5) and 2-cyclohepten-1-one (C7) in Canary Islands banana fields naturally infested with BW. 1,3-dimethoxybenzene (C5) significantly reduced the attraction of BW adults to sordidin (BW aggregation pheromone) in banana fields. C5 was detected in the field using GC-MS. C1 and C2 had a mild repellent effect influenced by seasonal changes. C7 VOC did not repel BW in the field. Site and season affected VOCs repellence to BW. Climate may influence VOCs evaporation and therefore their repellent efficacy. VOCs modify BW spatial ecology under field conditions. The inverse distance weighted (IDW) interpolation technique showed changes in BW infestation patterns after application of VOCs in the field. In conclusion, VOCs from biocontrol fungi reduced BW attraction to its aggregation pheromone in banana fields. These responses to experimental BW repellents were influenced by weather and BW population size. BW repellents have potential to be used in "push-pull" strategies to manage BW sustainably in banana crops.

## INTRODUCTION

Bananas (*Musa* spp.; Linnaeus, 1753) are essential for food security (*Nelson, Ploetz & Kepler, 2006*). Spain (Canary Islands) is the top European Union banana producer (*Food and Agriculture Organization (FAO), 2020*), with 420.144 tons per year (*Ministerio de Agricultura y Pesca, Alimentación y Medio Ambiente (MAPA), 2021a*). The main pests and pathogens that affect bananas are banana weevil (BW, *Cosmopolites sordidus*; Germar, 1824; Coleoptera: Curculionidae), plant-parasitic nematodes and the fungal pathogen, *Fusarium oxysporum* f. sp. cubense (Foc) ((Smith) Snyder and Hansen). The latter is the causal agent of Panama Disease (*Fusarium* wilt) (*Ostmark, 1974*; *Dubois et al., 2004*; *Oberprieler, Marvaldi & Anderson, 2007*; *Waweru et al., 2014*). These pests and pathogens harm banana roots and corm, reducing water and nutrient absorption (*Gold, Pena & Karamura, 2001*). This decreases banana vigour, size and may lead to plant death (*Nankinga & Moore, 2000*; *Dubois et al., 2004*; *Waweru et al., 2014*). BW can act as a Foc tropical race 4 vector strain, the main current threat to bananas worldwide (*Meldrum et al., 2013*). Therefore, control of BW may also reduce the spread of banana pathogens (*Guillén-Sanchez et al., 2022*).

*Cosmopolites sordidus* is the main pest of banana plantations in the Canary Islands (Spain) (*Ministerio de Agricultura y Pesca, Alimentación y Medio Ambiente (MAPA), 2016*). Infestations of banana fields by BW can lead to up 90% loss in yield (*Carballo, 1998*; *Musabyimana et al., 2001*; *Muñoz-Ruiz, 2007*). Larval feeding tunnels in banana corm cause plant damage (*Franzmann, 1972*). These tunnels alter nutrient content and plant stability (*Gold, Pena & Karamura, 2001*), delay flowering, weaken the plant and even destroy the root system. These effects cause a reduction of fruit size and production, as well as growth of offshoots plants and an increase in susceptibility to other pests and diseases (*Piedra-Buena Díaz et al., 2021*). In severe cases, this can lead to plant death (*Rukazambuga, Gold & Gowen, 1998*).

Due to its cryptic behaviour, BW is a difficult pest to control (*Tresson et al., 2021*). Few approaches can be used to manage this pest in an environmentally safe manner. Traps with BW aggregation pheromones (sordidin) are used usually by farmers to monitor BW populations (*Reddy, Cruz & Guerrero, 2009*). Chemical insecticides applications are implemented when large number of banana plants are infested. However, a limited number of chemical products (only oxamyl, fosthiazate, cyhalothrin and spinosad) are registered for BW control, since policies aim to minimize the use of toxic pesticides (*Ministerio de Agricultura y Pesca, Alimentación y Medio Ambiente (MAPA), 2021b*). Bananas are consumed fresh, and therefore insecticides are used only in exceptional cases. However, BW control by chemicals can be ineffective because BW is often hidden in the plant and leaf litter. The current limitation in the use of chemicals by the EU Green Deal policies implementation before 2030 (*Helepciuc & Todor, 2022*), has led banana farmers to introduce biocontrol organisms such as entomopathogenic nematodes (EPNs, *Steinerma* spp., *Heterorhabditis* spp., *etc.*) or endophytic biocontrol fungi (*Beauveria bassiana, Metarhizium anisopliae*) to control BW populations. Solarization and bio-fumigation (*Ministerio de Agricultura y Pesca, Alimentación y Medio Ambiente (MAPA), 2016*) are

also used in new commercial banana plantations. New solutions for controlling BW are needed. The use of volatile organic compounds (VOCs) as repellent compounds, together with aggregation traps, could be a solution to manage BW populations in the field.

VOCs with low molecular weight evaporate easily in the environment (*Hung, Lee & Bennett, 2015*). They are emitted into the atmosphere from natural marine and terrestrial sources (*Guenther et al., 1995*). VOCs play important roles in ecological and physiological processes of many organisms such as fungi (*Splivallo et al., 2011*; *Kramer & Abraham, 2012*). Fungal VOCs belong to diverse chemical groups such as monoterpenoids, sesquiterpenes, alcohols, aldehydes, aromatic compounds, esters, furans, hydrocarbons, and ketones (*Splivallo et al., 2007*; *Campos, de Pinho & Freire, 2010*; *Kramer & Abraham, 2012*). Several VOCs are important semiochemicals that affect fungus-insect interactions (*Yanagawa, Yokohari & Shimizu, 2009*). They may function as attractants and/or repellents for insects and other invertebrates. Among other important roles, they can modify insect feeding behaviour, elicit mating and oviposition in insects, and signal suitable habitats (*Davis et al., 2013*). Insect antennae and other parts in their body (legs) have chemoreceptors capable of detecting VOCs. Therefore, VOCs have a potential role for sustainable management of BW by modifying its behaviour and its ability to search for crop hosts (*Musa* spp.). VOCs isolated from *Penicillium expansum* (Link ex. Thom.), such as styrene and 3-methylanisole, have repellent effects over pine weevils (*Hylobius abietis* (L.)) in the laboratory (*Azeem et al., 2013*). Methyl salicylate produced by filamentous fungi and yeast isolated from faeces of pine weevil females affects host-odor interaction with *Pinus sylvestris*, reducing pine weevil attraction to VOCs of the tree (*Azeem et al., 2015*).

Several VOCs from nematophagous and entomopathogenic fungi, such as 1,3-dimethoxybenzene and 2-cyclohepten-1-one, isomer of 3-cyclohepten-1-one, have shown repellent activity to BW in the laboratory (*Lozano-Soria et al., 2020*; *Mestre-Tomás et al., 2023*; *Lozano-Soria et al., 2024*). In this study, we evaluated a set of these VOCs in commercial banana field plantations in the Canary Islands (Spain) with natural infestations of BW.

## MATERIALS AND METHODS

Portions of this text were previously published as a part of a thesis (*Lozano-Soria, 2023*).

### Volatile organic compounds and pheromones

We have used volatiles from biocontrol fungi (C1, C2, C5 and C7). C1 (styrene) and C2 (benzothiazole) were produced by the isolates Bb203 and Bb1TS11 form the entomopathogenic fungi *Beauveria bassiana* (*Lozano-Soria et al., 2020*, *2024*). C5 (1-octen-3-ol) was produce by the isolate Mr 4TS04 of the entomopathogenic *fungus Metarhizium robertsii* and the isolate *Pc123* of the nematophagous fungus *P. chlamydosporia* (*Lozano-Soria et al., 2020*, *2024*). C7 (cycloheptene-1-one) was identified from the entomopathogenic and nematophagous fungi (*Lozano-Soria et al., 2020*, *2024*). All the strains are preserved in the Spanish Type Culture Collection (CECT) (Table S1). These VOCs were formulated by mixing them individually (500 µL) with silica

gel (2 g) (60A, 70–200 μ; CARLO ERBA Reagents S.A.S, Cornaredo, Italy). Formulations were placed in miracloth article (Calbiochem Millipore Corp., Burlington, MA, USA) envelopes (3.5 × 2.5 cm) (Fig. S1B). These were sealed and stored at 4 °C in the dark in closed zip bags enclosed in vacuumed bags before use. Cosmolure™ (P160-Lure90; Chemtica International S.A., Costa Rica) or Ecosordidina30 (ECOBERTURA S.A, Canary Islands, Spain), both commercial BW aggregation pheromone formulations with an expected field efficacy of 90 and 30 days, respectively, were also used in experiments (Figs. S1A, S1E).

## Banana weevil traps

Chemicals (VOCs and pheromones) were placed in open dispensers in pitfall traps (Scyll Agro®, Hastingues, France) (*Piedra-Buena Díaz et al., 2021*). They consist of an empty vessel buried in the ground that contains the active compound (pheromones and/or VOCs) and a lid enclosing the trap, leaving a space through which the weevils fall into the trap (Figs. S1C, S1D).

## Field experiments

Seven field experiments were carried out in the Canary Islands between April 2019 and December 2021. Four treatments were used in the traps: (i) pheromone only, (ii) pheromone + VOC, (iii) VOC, and (iv) empty traps as a control to estimate natural captures (no chemicals; Table 1). Traps were placed in commercial banana fields in North Tenerife (Canary Islands, Spain), mostly with high natural BW infestations identified previously (Fig. 1). Traps with treatments were placed randomly within each field. Traps were set ten meters apart in rows. Distance between adjacent rows was 18 m, generating a trap network covering the area of the field. The total number of traps (N) ranged from 18 to 36 traps/field depending on the size of the field (Table 1). BW adults captured in the traps were periodically counted (7–14 days) and removed to avoid reintroduction of insects in the field. VOCs were replaced every 3 weeks to maintain their efficacy (Table 1). Besides, cumulative (summation of BW captured in all time points) captures per trap were calculated by adding all BW scored during the whole experimental period.

## Volatile organic compound analysis

Tenax TA fritted glass TD tubes were used to detect VOCs in the field (*Cai et al., 2015*). Seven tenax TA fritted glass TD tubes (17.8 cm length × 6 mm OD × 4 mm ID; 60–80 mesh; 60 mm/180 mg; Supelco, Bellafonte, PA, USA) were used for capturing VOCs in the field. Prior to use, they were conditioned (GERSTEL TC) and sent to the Canary Islands. They were then placed at the "Los Llanos" (LL) site (Fig. 1) one tenax next to one trap baited with C5 another close to a C5+pheromone baited trap and a further one close to an empty trap. They were left for 24 h at 0, 10 and 25 cm from traps, except for the empty trap where only one tube was placed at the closest distance (Fig. S2). After 24 h capturing VOCs, Tenax fibers were sent to Alicante and placed into the Gases Mases Spectrophotometry Service in the Common Research Facilities at University of Alicante. Them, tenax fibers were desorbed in a Thermal Desorption System (Gerstel TDS-2), from 40 °C (0.5 min) to 250 °C
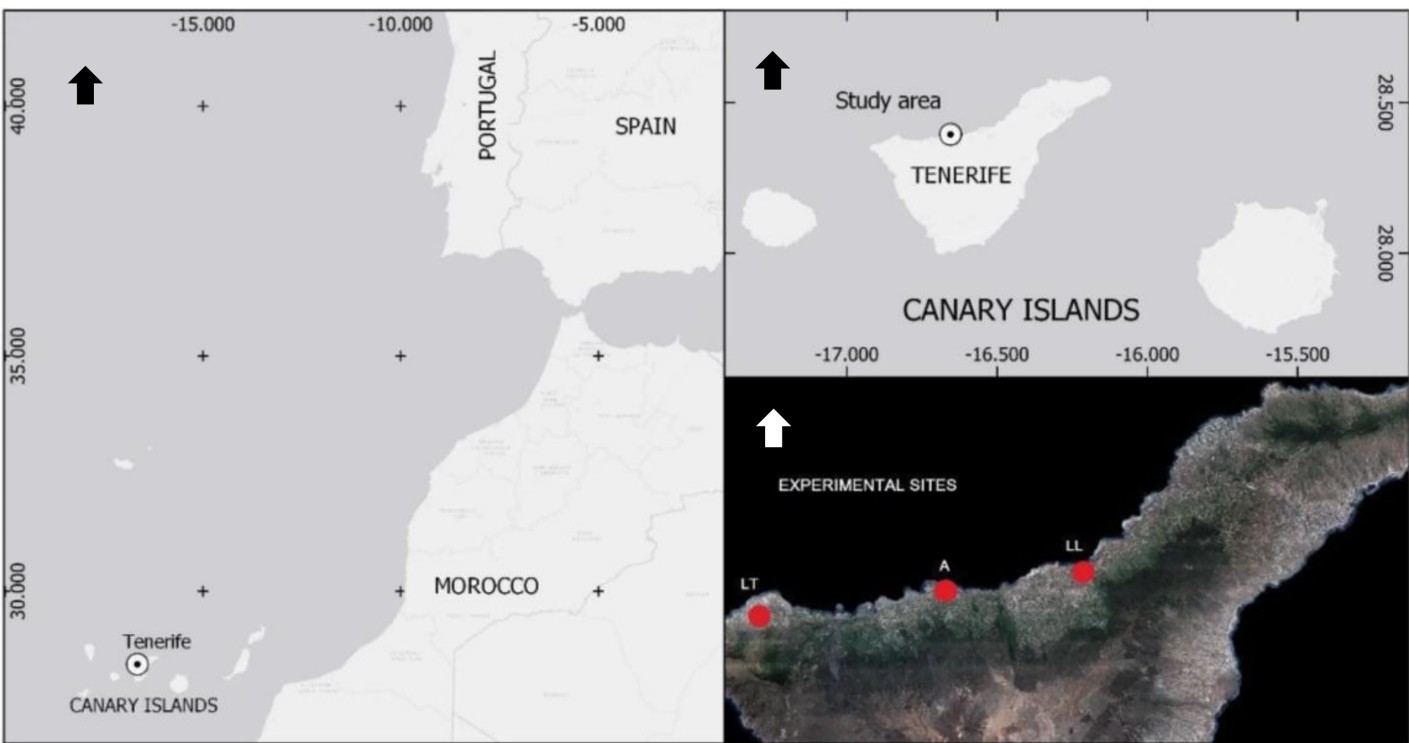

**Figure 1** **Geographical location and distribution of commercial banana fields naturally infested with BW selected for experiments.** All experiments were conducted in northern Tenerife (Canary Islands, Spain). Sites were in Las Toscas (LT, Buenavista del Norte), Alcaravanes (A, San Juan de la Rambla) and Los Llanos (LL, La Orotava). Arrows point north.

(60 °C/min, 6 min). VOCs were then injected in a Gas Chromatograph coupled to a Mass Spectrometer (Agilent model 6890N gas chromatograph-Agilent 5973 Network Mass Spectrometer; column: DB-624 30 m, 0.25 mm ID 1.4 m, J&W Scientific, Folsom, CA, USA). The chromatography program used had an initial temperature of 35 °C (5 min) and a 3 °C/min increasing curve until 150 °C. Afterwards, a 5 °C/min increasing curve was used until 250 °C. Total analysis time was 64.33 min. The electron impact ionization was 70eV at 230 °C. A single quadrupole was used as an analyzer at 150 °C in the Mass Spectrometer working in SCAN mode (range of 15–350 m/z) as a detector. Wiley275 library was used for identifying VOCs.

## Meteorology data

Meteorological data were collected from El Drago, Buenavista del Norte and La Orotava meteorological stations (*Agrocabildo, 2015*). Daily measurements of mean temperature (°C), mean relative humidity (%) and precipitation (mm) were used for the analysis. Mean meteorological data were plotted weekly.

## Banana weevil trap capture interpolation maps

Surface maps were plotted from BW captures over time using Inverse Distance Weighted (IDW) interpolation with QGIS Software (www.qgis.org). QGIS generates an IDW interpolation of a point vector layer (insect trap data). IDW estimates unknown values by

**Table 1 VOC assays in fields naturally infested with *Cosmopolites sordidus*.**

| Experiment number | VOC | N° traps/treatment | Duration (weeks) | VOC replacement | BW monitoring | Farm location (GPS coordinates) | Age of the field |
|---|---|---|---|---|---|---|---|
| 1 | C1 and C2 | pheromone (n = 10); pheromone+C1 (n = 10); pheromone+C2 (n = 9) | 4 (April 16th–May 22nd, 2019) | – | Weekly | "Alcaravanes" San Juan de la Rambla (28°23'39.58"N, 16°39'19.04"W) | More than 20 years old |
| 2 | | Ecosordidina30 pheromone N Total Traps = 29 | 6 (June 4th–July 16th, 2019) | 3rd week (50% traps) | | | |
| 3 | | | 12 (October 16th, 2019–January 10th, 2020) | 4th week (50% traps) | Bi-weekly | | |
| 4 | C5 | Pheromone (n = 10); pheromone+C5 (n = 10) Ecosordidina30 pheromone N Total Traps = 20 | 12 (October 16th, 2019–January 10th, 2020) | 4th week (50% traps) | Bi-weekly | "Las Toscas" Buenavista del Norte (28°22'19.94"N, 16°50'36.20"W) | More than 20 years old |
| 7 | | Pheromone (n = 9); pheromone+C5 (n = 9); VOC C5 (n = 9), empty traps (n = 9) Cosmolure™ pheromone N Total Traps = 36 | 9 (October 7th–December 9th, 2021) | 3rd week (100% traps) | Weekly | "Los Llanos" La Orotava (28°24'39.8"N, 16°30'57.0"W) | More than 20 years old |
| 5 | C7 | Pheromone (n = 10); pheromone+C7 (n = 10) Ecosordidina30 pheromone N Total Traps = 20 | 12 (October 16th, 2019–January 10th, 2020) | 4th week (50% traps) | Bi-weekly | "Las Toscas" Buenavista del Norte Greenhouse farm (28°22'19.94"N, 16°50'36.20'W) | 1–2 years old |
| 6 | | Pheromone (n = 9); pheromone+C7 (n = 9) Cosmolure™ pheromone N Total Traps = 18 | 11 (June 23rd–September 8th, 2021) | 3rd and 6th weeks (100% traps) | Weekly | "Los Llanos" La Orotava (28°24'39.8"N, 16°30'57.0"W) | More than 20 years old |

Note:
All experiments used silica gel for dispensing VOCs except Exp.3 (Ecosordidina30 matrix). n = traps/treatment. C1–C7: see Table S1 for VOCs identities.

averaging the values of sample data points in the vicinity. The closer the sampled point (known value) to the estimated point (unknown value), the more influence it has in the process. It assumes that the variable being estimated decreases in influence with distance from its sampled location. IDW uses spatial autocorrelation and is a quick deterministic interpolator (*Li & Heap, 2011*). IDW is an exact interpolator, where the maximum and minimum values in the interpolated map can only occur at sample points. IDW interpolations can generate predictions even when pest presence is low (*Cohen et al., 2022*).

## Statistical analysis

One-way ANOVA followed by Tukey's HSD was used to analyze differences between treatments (fixed factor) in cumulative values of BW captures per trap for the entire experimental period. Cumulative values of BW captures per trap/treatment were

calculated by successively adding (summing) the BW captures/trap found in all samples taken in each field experiment. Prior to ANOVA, homogeneity of variances was checked using the Levene test. The normality of the distribution of the residuals in the models was checked using the Shapiro test. If the homogeneity of variance of the data or the normality of the residuals of the ANOVA model were not met, the data were transformed using the square root or logarithms. When transformed data did not meet the parametric assumptions, the Kruskal–Wallis non-parametric test was used.

A negative binomial generalized linear mixed model fit by maximum likelihood (Laplace approximation; GLMM: 'glmer.nb' function from the 'lme4' package; *Bates et al., 2015*) was used to estimate fixed and random effects in the response of BW captured in traps. As traps were always the same, only the volatiles (repellents and sordidin) were replaced, a repeated measures model was used on the same trap, with trap considered as a random effect and treatment and time as fixed factors in the model. *Post-hoc* analyses were also performed to identify differences in response variables between treatments at each monitoring time. Pairwise comparisons with estimated marginal means (emmeans function and package; *Lenth, 2022*) were performed using Sidak's HSD test for the GLMM data. Diagnostics of the models were examined using the DhARMa package (*Hartig, 2022*). To analyze the relationship between BW captures and each meteorological variable, Pearson's correlation coefficients were calculated for each experiment. The normality distribution of the data was checked with Shapiro test. In case that normality distribution of the data was not met, we performed Spearman's test. All statistical analyses and graphs were performed in R version 3.6.1 (*R Core Team, 2021*) and using GraphPad Prism version 6.01 (*GraphPad, 2021*). The data analysis workflow is shown in Fig. S3.

## RESULTS

### Fungal volatile organic compounds performance vary in their repellence to banana weevil under field conditions

The most repellent effect was observed for 1,3-dimethoxybenzene (C5) in the "Las Toscas" experiment (Fig. 2). In this experiment (Experiment 4, Table 1), the presence of VOC C5 in traps together with the pheromone Cosmolure$^{TM}$ significantly reduced the cumulative value of BW adults captured per trap (26.8 ± 16.752) compared to traps in which the pheromone was applied alone (14.6 ± 7.106) (one-way ANOVA, F = 4.495, DF = 1.18, $p$ = 0.048; Fig. 2A). In addition, cumulative values of BW adults captured per trap with C5 only in the "Los Llanos" experiment (Experiment 7) were significantly (one-way ANOVA, F-value = 66.02, $p$-value < 0.0001) lower (7.22 ± 5.23) than those in traps with pheromone only (100.77 ± 37.48) and, also with pheromone and C5 together (97.55 ± 22.75) (Fig. 2B). We detected, under field conditions, C5 VOC in the air close to traps baited with the volatile (Table S2). As expected, C5 amount decreased with distance from traps.

Styrene (C1) and benzothiazole (C2) baited traps (Fig. S4) caused a reduction tendency in BW cumulative captures in Experiments 2 (33.00 ± 22.80 and 36.66 ± 23.53) and 3 (41 ± 32.59 and 34.77 ± 26.38) compared to those found in pheromone baited traps (39.3 ± 25.47 and 44.4 ± 31.75). However, these cumulative captures were not significantly

different from those in pheromone only containing traps (Figs. S4B and S4C). 2-cyclohepten-1-one (C7) showed no effect on BW captures with either low or high BW population levels in the field (Experiment 5 and 6) (Figs. S4D, S4E).

## Site and season affect volatile organic compounds repellence to banana weevil

Number of BW adults captured per trap weekly or biweekly was highly variable. Experiment 1 where VOCs (C1 and C2) were not replaced indicated a 3-week period of VOCs activity (Figs. S5 and S6). This was reflected in a tendency of increasing capture in traps with VOCs (C1 or C2) only form third weeks onwards. Therefore, VOCs were replaced after 3–4 weeks for the rest of the experiments (Table 1).

VOC C5 in "Las Toscas" Experiment 4 reduces BW captures in traps containing both VOC and BW pheromone (2.43 ± 2.061) compared to traps with BW pheromone only (4.47 ± 4.534) (Fig. 3). No significant differences were found in treatment:time interaction (Fig. 3). However, significant differences were found in factor treatment and factor time. Pheromone traps only significantly attracted more BWs than traps containing both active compounds (GLMM Negative Binomial, Estimate = 1.0046, z-value = 3.005, $p$-value = 0.00225; Table S3, Diagram S1 and S2) than traps containing both active compounds. The winter period coincided with a reduction in temperature and precipitation (Fig. S7). The variance of the trap was 0.1412.

In the second assay performed with VOC C5 ("Los Llanos", Experiment 7), empty traps and traps with C5 only were also included. The number of BWs was significantly lower in traps with fungal VOC C5 only than in traps with pheromone only (GLMM Negative Binomial, Estimate = 1.917, z-value = 5.036, $p$-value < 0.001; Fig. 4, Table S3, Diagram S1 and S2). In this experiment, meteorological events (Fig. S8) conditioned field results. For instance, a mild precipitation in the third week resulted in a significant increase in BW captures (Fig. 4). There was no significant difference in the number of BWs caught across the treatments (GLMM Negative Binomial Estimate 0.71488; z-value = 1.691, $p$-value = 0.0908). A high precipitation occurred at the end of the experiment. This may have affected trap captures since some of them were filled with rainwater. Capture in the last weeks were consequently lower than in the previous one (Fig. 4). The variance of the trap capture rate is 0.05187.

No significant differences were found in BW trap captures for C1 and C2 in Experiments 2 and 3 for treatment factor nor for treatment:time interaction (Figs. S9–S12). C7 was first evaluated in "Las Toscas", Experiment 5 (Table 1). BW captures were extremely low (2–3 individual per trap) for the whole experiment and no significant differences were found for treatment, time, and the interaction (Fig. S13). C7 was also tested at "La Orotava", Experiment 6, during summer months (Figs. S14, S15). In this case only time show significant reduction in BW during summer. No differences were found for treatment factor nor for the treatment:time interaction.
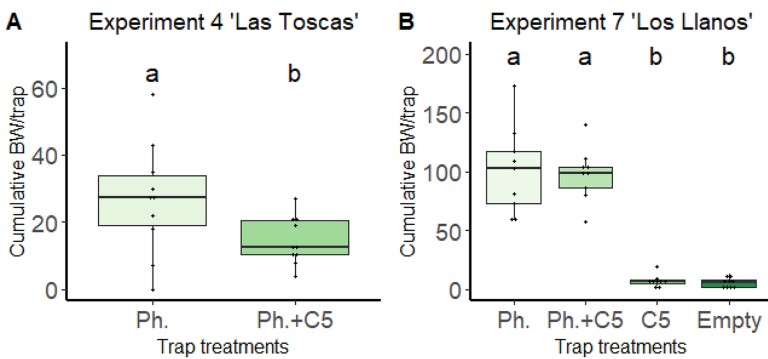

**Figure 2 Effect of VOC C5 on BW adult trapping in banana fields from Canary Islands.**
(A) Experiment 4 "Las Toscas". Pheromone, Pheromone + VOC C5. Unpaired t-student, two tailed;
$p$-value = 0.048 ($n$ = 10). (B) Experiment 7 "Los Llanos". Pheromone, Pheromone + VOC C5, VOC C5,
Empty Traps. One-way ANOVA; $F_{3,32}$ = 66.02; $p$-value < 0.0001 ($n$ = 9). Columns represent total BWs
captured over time per trap/n. Error bars = SEM. BW = Banana weevil (*C. sordidus*); Ph. = BW pher-
omone; C5 = 1,3-dimethoxybenzene. Lowercase letters indicate significant difference between treat-
ments.                                         

## VOC 1,3 dimetoxybenzene modifies banana weevil spatial ecology under field conditions

BW shows an aggregate distribution in the field. Surface model maps are different at the
start of the experiments from those at the end (Figs. 5, S16 and S17). Videos showing time
lapses of BW captures are included as GIF files (Figs. S18–S20). They reveal areas with high
(<45 BW/trap/week) and low ($\leq$5 BW/trap/week) BW density. IDW interpolations
indicated the areas of occurrence and movement of the BWs in the field.

In "Las Toscas" (Experiment 4) (Table 1) with fungal VOC C5, there were five initial
BW foci (outbreaks) across the field (Fig. 5A). They were associated with pheromone
baited traps (3) and C5+Pheromone traps (2). BW field populations decreased with time
(Fig. S18, Top). Foci associated with C5+pheromone completely disappear (Fig. S18, Top).
At the end of the experiment, two new emerging foci, characterized by middle values of
BW captures per trap, appear in pheromone only containing traps (Fig. 5B). "Los Llanos"
C5 (Experiment 7) started with three main foci (associated to pheromone baited traps) in
the south part of the field plus three further foci in the northern part in this case associated
with pheromone+C5 baited traps (Fig. 5C). BW then spread throughout the field (Fig. S18,
Bottom). At the end the experiment, BW population decreased, but the initial three foci,
associated with pheromone baited trap, now with lower captures, remained (Fig. 5D). The
other three initial foci with C5+pheromone completely disappear (Fig. S18, Bottom).

In the C1–C2 "Alcaravanes" (Experiment 1), there were two early BW outbreaks in the
northern area of the field (associated with pheromone+C1 baited traps) and a main one in
the south-east part close to pheromone baited trap (Fig. S16A). In the subsequent weeks,
the south focus increased and moved to the east, to a Pheromone+C1 baited trap (Fig. S19,
Top). At the end only two emerging foci in pheromone traps were observed (Fig. S16B).
Regarding the C1–C2 (Experiment 2), performed in early summer, BW was less present
compared to prior experiments (Fig. S16C). However, foci shifted with time (Figs. S16D

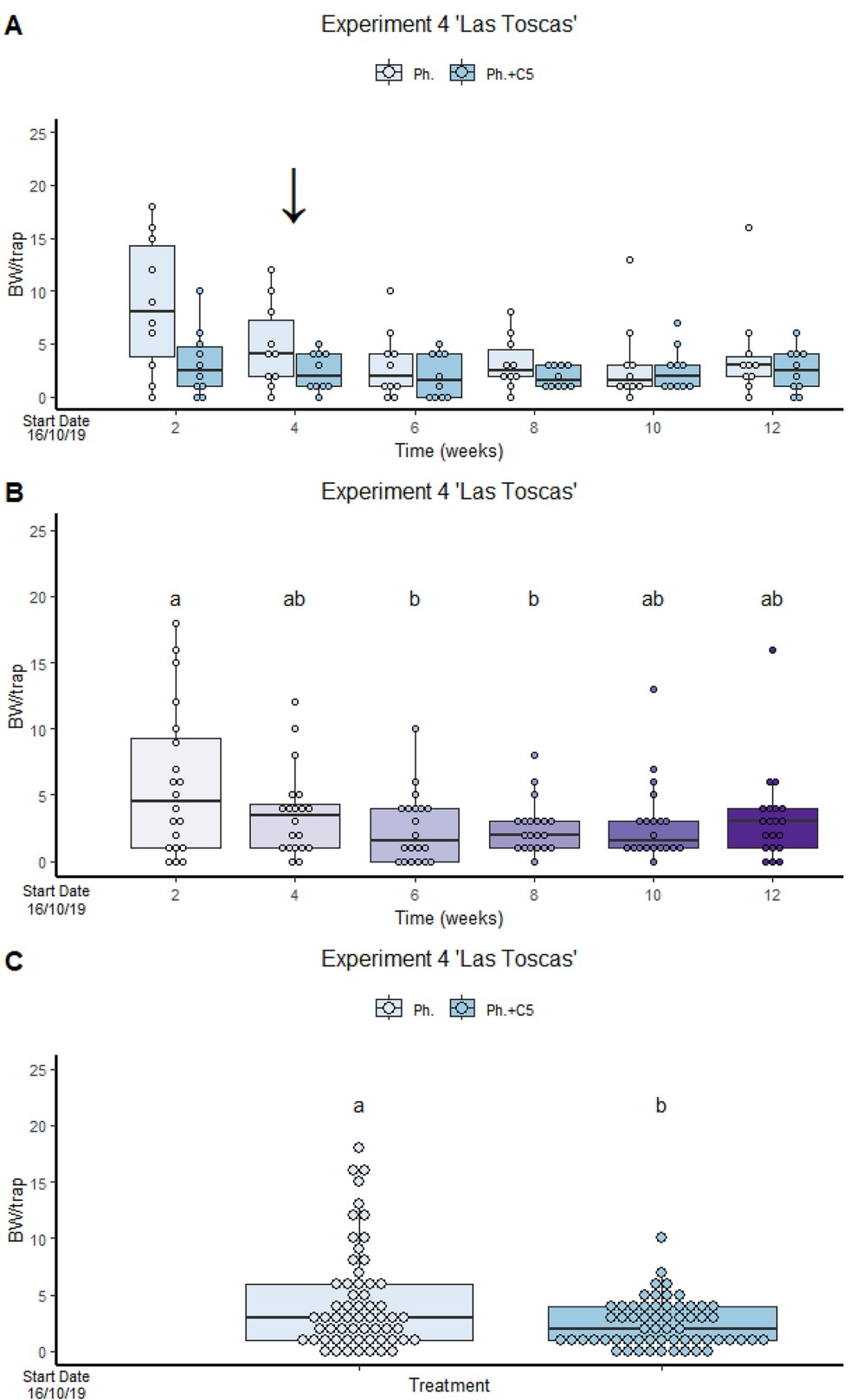

**Figure 3 VOC C5 in "Las Toscas" Experiment 4 reduces BW captures in traps containing both VOC and BW pheromone.** (A) *Cosmopolites sordidus* captures per treatment with time. (B) *C. sordidus*

**Figure 3 (continued)**
captures in each monitoring time. (C) *C. sordidus* captures per treatment. Treatments: C5 with pheromone and pheromone only traps. Columns represent BWs trapped/treatment each monitoring time. Error bars = SEM. Arrows indicate dates of fungal VOC dispensers' replacement (see Table 1). Letters indicate significant differences (emmeans analysis). Lowercase letters indicate significant difference between treatments.

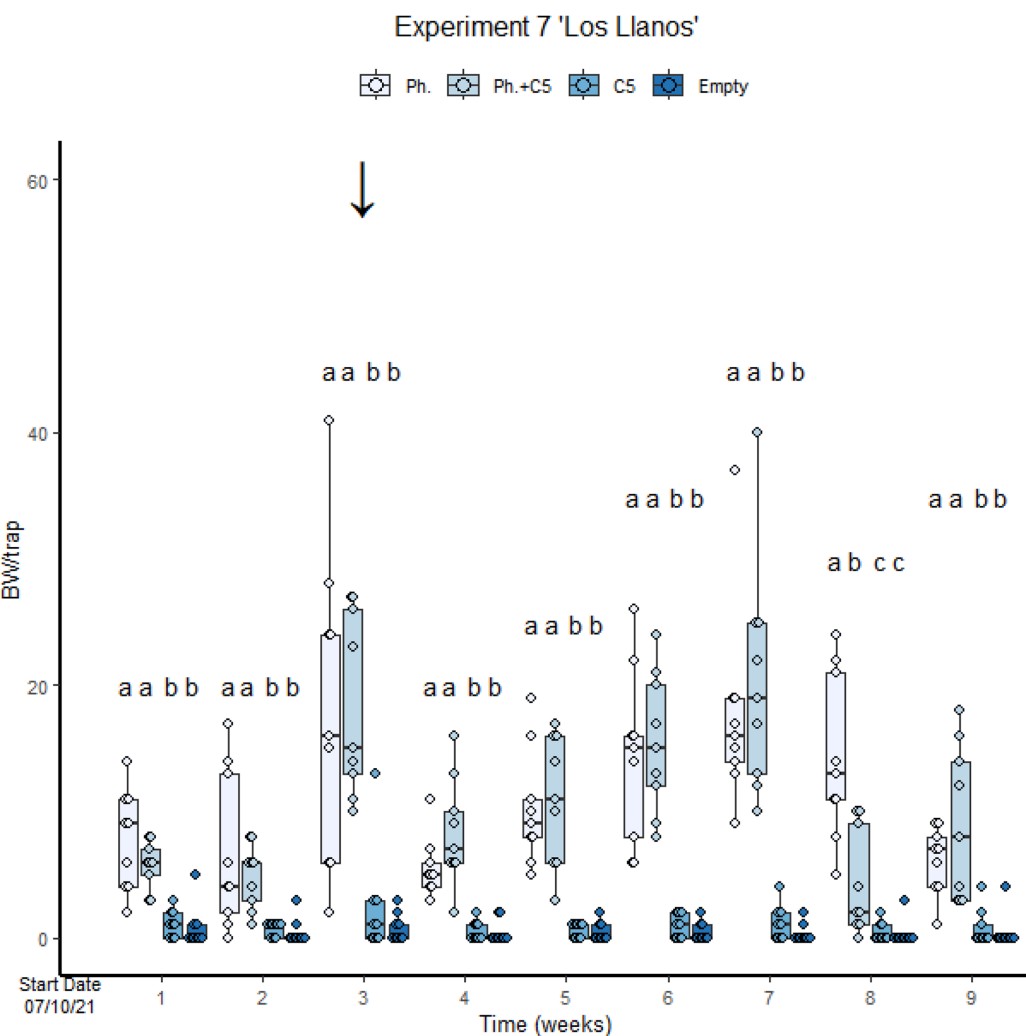

**Figure 4** *Cosmopolites sordidus* **captures per treatment with time in Experiment 7 "Los Llanos".** Treatments: C5 and pheromone alone and in combination and empty traps. Columns represent BWs trapped/treatment each monitoring time. Error bars = SEM. Arrows indicate dates of fungal VOC dispensers' replacement (see Table 1). Letters indicate differences between treatment in each monitoring times (emmeans analysis). Lowercase letters indicate significant difference between treatments.

and S19, Middle). In the last experiment in "Alcaravanes" (Experiment 3) two big BW outbreaks associated with pheromone+C2 and pheromone baited traps in the south-west area of the field (Fig. S16E) decreased and moved with time (Fig. S19, Bottom). At the end

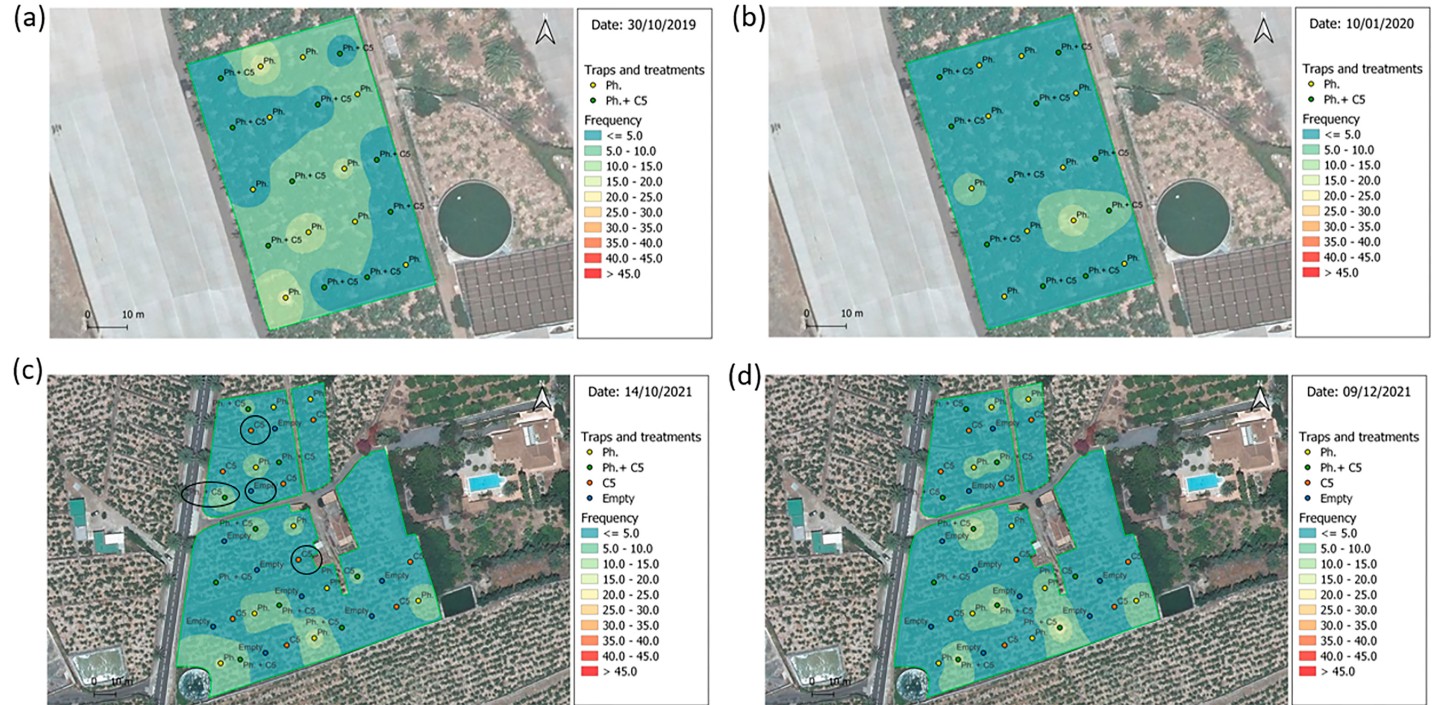

**Figure 5** **Spatial distribution of *Cosmopolites sordidus* adults.** Weekly captures from the first (left) and the last (right) monitoring times in banana field shown by surface maps constructed based on the Inverse Distance Weight (IDW) interpolatio. (A and B) Experiment 4 "Las Toscas". Yellow dots = location of pheromone traps in the field; Green dots = location of traps with pheromone and VOC C5 in the field. (C and D) Experiment 7 "Los Llanos". Yellow dots = location of pheromone traps in the field; Green dots = location of traps with pheromone and VOC C5 in the field; Orange dots = location of C5 traps in the field; Blue dots = location of empty traps in the field. Blue colors = Low density; Red colors = High density; Black ovals = traps used for C5 GC-MS analysis.

of the experiment, in the middle of the winter, BW was almost absent from traps (Fig. S16F).

In "Las Toscas" C7 (Experiment 5) (mesh-covered field) BW was scarcely present (Figs. S17A, S17B). The geostatistical model did not perform well with such low BW captures (Fig. S20, Top). In C7 "Los Llanos" (Experiment 6) there was a large initial BW outbreak in the middle of the field close to a fungal VOC C7+pheromone and pheromone baited traps (Fig. S17C). In the next weeks, the big outbreak split into two little foci areas at the south-east part of the field (Fig. S20, Bottom) finally associated with pheromone traps (Fig. S17D).

## DISCUSSION

Styrene (C1), benzothiazole (C2), 1,3-dimethoxybenzene (C5) and 2-cyclohepten-1-one (C7), VOCs from biocontrol fungi *B. bassiana*, *M. robertsii* and *P. chlamydosporia* repel BW under laboratory conditions (*Lozano-Soria et al., 2020*). In this work, we show that 1,3-dimethoxybenzene (C5) can disturb BW pheromone sordidin attraction to BW in the field. This VOC is emitted by a *M. robertsii* strain native from banana fields in the Canary Island (Spain). C5 is also produced by the nematophagous biocontrol fungus *P. chlamydosporia* (*Lozano-Soria et al., 2020*; *Mestre-Tomás et al., 2023*). Other studies also

showed that 1,3-dimethoxybenzene is also generated by *Aspergillus versicolor* (*Matysik, Herbarth & Mueller, 2009*). A C5 isomer (1,4-dimethoxybenzene) produced by a *P. chlamydosporia* strain has been described as attractive and toxic to the nematode *Meloidogyne incognita* juveniles (J2) and inhibits *M. incognita* egg-hatching (*Pacheco et al., 2022*).

It is somehow puzzling that 2-cycloheptanone (commercially available used instead of 3-cycloheptanone), which repelled BW the most in olfactometers (*Lozano-Soria et al., 2020*), was not active under field conditions. C7 is a common emitted fungal volatile. *Agaricus bisporus* and *Aspergillus candidus* produce this VOC during growth (*Fischer et al., 1999*; *Aisala et al., 2019*; *Feng et al., 2021*). Besides, some Mucorales, like *Mucor plumbeus* or *Mucor racemosus*, also produce 3-Cycloheptanone during growth and development. These fungi also have the enzymatic machinery to reduce this ketone into an alcohol (3-cycloheptanol; *Lemière et al., 1975*).

The other two VOCs tested in this work, Styrene (C1) and benzothiazole (C2), varied in their BW repellency in banana plantations, which was significantly lower than in the laboratory. Styrene (C1) and benzothiazole (C2) have been described as effective repellents to the red palm weevil (*Rynchophorus ferrugineus; Jalinas et al., 2022*). Under laboratory conditions styrene (C1), has been also isolated from *Penicillium expansum* in the feces and frass of the pine weevil (*H. abietis*), reducing attraction of adults to pieces of Scot pine twigs (*Azeem et al., 2013*). VOCs from selected *B. bassiana* and *M. anisopliae* strains have repellent effects on the granary weevil (*Sitophilus granarius*) (*Selitskaya et al., 2016*; *Mitina, Selitskaya & Schenikova, 2020*). A further strain of *B. bassiana* is repellent to the western flower thrips (*Mitina, Stepanycheva & Petrova, 2019*).

Multiple factors influence insect trapping such as density of individuals, temperature, moonlight, traps, lure placement and maintenance and immigration of individuals (*Rannestad, Sæthre & Maerere, 2011*). Field trials for BW control are affected by climate factors (*Bakaze et al., 2022*). In our study, site, weather, and season affected VOCs repellence to BW. Experiments conducted during autumn and early winter had higher density of BW due to the high relative humidity and cooler temperatures which favour insect mobility (*Bakaze et al., 2022*). Therefore, environmental factors, mainly rain and humidity, can alter trap captures generating variability and then affecting the robustness of the results obtained (*Okolle et al., 2020*). Weather, mainly temperature, modulates BW behaviour, it can also modify the volatility of the organic compounds tested and then its perception by BW (*Bakaze et al., 2020*, *2022*). Most VOCs evaluated in our work modify BW spatial ecology under field conditions. In all our field experiments with volatiles BW foci ended closed to pheromone baited traps or completely disappeared. VOCs either confuse insects (no BW accumulation/foci in traps) or they push BW to pheromone containing traps. This supports the view that fungal VOCs tested push BW pooling them in the sordidin baited traps.

Under low BW populations, management is acceptable, but when population increase, the efficacy of the control methods declines (*Miller, 2002*; *El-Sayed et al., 2006*; *Audley et al., 2022*). This has been found in other VOCs experiments (*Agnello et al., 2021*). BW

mobility may also affect trapping. To this respect there are disperser and non-disperser phenotypes of BW (*Carval et al., 2015*).

The distance of the repellent-containing traps from each other and from the attractant-containing traps must also be considered as a crucial factor in the distribution of the traps and treatments in the field for the design of push-pull strategies (*Rivera et al., 2020*). Distance affects VOCs repellent concentration and the insect ability to detect them. In this work, fungal VOC C5 was detected in the field at 0.25 m from the delivery source, even in traps with the VOC in combination with the pheromone. Other studies have found that the distance of repellents affecting captures in pheromone traps is less than a meter in most cases (*Byers et al., 2018*, *2020*; *Rivera et al., 2020*).

The interpolation models here presented create effective predictions of pest abundance that can be used by growers to compare across regions of the fields. However, the performance of the models can vary within seasons (*Cohen et al., 2022*). Surface maps can be used to localize the insects in the field and to show the spatial distribution during the establishment of the pest (*Weber et al., 2018*). They can locate the pest in the field and help with trap placement. Newly detached banana suckers are highly susceptible to BW (*Bakaze et al., 2022*). In our study traps with C5 significantly reduce BW captures. Then, C5 could therefore be used to prevent BW adults from infesting new banana plantations. Alternatively, insects could be pushed out from fields with BW infested adult banana plants. They could then be mass captured with sordidin and host kairomone traps in a push and pull strategy.

Most investigations are currently conducted about the biological properties of VOCs as repellent to modify the behaviour of pests and pathogens under controlled conditions. However, fewer are performed in the field to prove the results that are achieved *in vitro* (*Okolle et al., 2020*). VOCs from fungi may act as recognition signals for beetles to maintain specific microbial communities and to respond to environmental stimuli (*Kandasamy et al., 2019*). On the contrary, in our study, recognition of VOCs from entomopathogenic fungi as repellents by BW may have evolved as a defence mechanism against insect pathogens. We are developing this biological communication strategy between banana weevils and biocontrol agents to obtain an efficient repellent with proven efficacy in the field.

## CONCLUSION

In conclusion, C5 significantly repels BW. This effect is dependent on BW population density levels. We confirm C5 field detection in the vicinity of the traps containing C5. This means that BWs could sense this VOC and distinguish it from empty traps. It is also detected in traps with both pheromone and C5 dispensers, so the volatility is sufficient to be detected considering the field environment (banana plants). BW prefers comfortable temperature (20–25 °C), darkness and has negative phototropism. In springtime days became longer (more light hours per day) than in winter and this reduces BW activity. The lack of moisture also reduces BW development (*Gold, Pena & Karamura, 2001*).

C5 could also be used with other fungal VOCs repellents or volatiles, such as ethyl acetate. Ethyl acetate is used with ferruginol increasing *Rhynchophorus ferrugineus*

captures (*Faleiro, 2006*; *Venugopal & Subaharan, 2019*). Likewise, with the *Odoiporus longicollis* aggregation pheromone and the host plant extract, the combination of both attracted more banana pseudostem weevils (*Palanichamy et al., 2011*). Adding Me-JA and 1-hexanol independently with banana plant extracts also increase pseudostem weevil captures (*Palanichamy, Boopathi & Uma, 2021*).

In addition, it would be interesting to evaluate VOCs with slow-release matrices and with higher doses to see if it makes any difference in the behaviour of BW. In a push-pull strategy it is a rather arduous task to achieve the correct repellent dosage rate in combination with the attractant. Therefore, this could set a new approach for better management of BW which has no effective control method for managing the populations in the field. This article provides evidence that BW repellents, previously unavailable, could then be a new sustainable tool for BW integrated pest management in the field.

## ACKNOWLEDGEMENTS

We thank ECOBERTURA S.A for providing the pheromones used in the field. Authors would like to thank members of the Plant Pathology Laboratory of the University of Alicante for their help and support. We would like to thank the farm staff: Víctor Martín (Coisba), Felix Soriano Benitez de Lugo and Melchor de Ponte (SAT FAST). We also thank technical support by Sagrario Lopez (Coisba), Diego Pérez, Miguel Chávez (SAT FAST), José Oramas and Ana García Carrión.

### Funding

This study was supported by funds from H2020 European Project number 727624, Microbial uptakes for sustainable management of major banana pests and diseases (MUSA) and by the PID2020-119734RB-I00 from the Spanish Ministry of Science and Innovation. There was no additional external funding received for this study. The funders had no role in study design, data collection and analysis, decision to publish, or preparation of the manuscript.

### Grant Disclosures

The following grant information was disclosed by the authors:
H2020 European: 727624.
Microbial Uptakes for Sustainable Management of Major Banana Pests and Diseases (MUSA).
Spanish Ministry of Science and Innovation: PID2020-119734RB-I00.

### Competing Interests

The authors declare that they have no competing interests.

### Author Contributions

- Ana Lozano-Soria performed the experiments, analyzed the data, prepared figures and/or tables, and approved the final draft.

- Ana Piedra-Buena Diaz performed the experiments, authored or reviewed drafts of the article, and approved the final draft.
- Federico Lopez-Moya conceived and designed the experiments, analyzed the data, authored or reviewed drafts of the article, and approved the final draft.
- Miguel Valverde-Urrea analyzed the data, prepared figures and/or tables, and approved the final draft.
- Jose J. Zubcoff analyzed the data, authored or reviewed drafts of the article, and approved the final draft.
- Jose Emilio Martinez-Perez analyzed the data, prepared figures and/or tables, and approved the final draft.
- Javier Lopez-Cepero conceived and designed the experiments, performed the experiments, authored or reviewed drafts of the article, and approved the final draft.
- Luis V. Lopez-Llorca conceived and designed the experiments, authored or reviewed drafts of the article, and approved the final draft.

## Data Availability

The data is available in the Supplemental Files.

## Supplemental Information

Supplemental information for this article can be found online at http://dx.doi.org/10.7717/peerj.19414#supplemental-information.

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
