# Peer review of "Volatile organic compounds from entomopathogenic and nematophagous fungi repel banana weevil (*Cosmopolites sordidus*) under banana field conditions"

_PeerJ, doi:10.7717/peerj.19414_

## Round 0.1 · original submission · Major Revisions

Dear authors,

Despite the criticism made by the reviewers, I have decided to allow you to answer their questions. Please carefully revise your paper tackling all the issues raised.

Reviewer 1 ·

Basic reporting

The manuscript “Volatile organic compounds from entomopathogenic and nematophagous fungi repel banana weevil (Cosmopolites sordidus) under banana field conditions” by Ana Lozano-Soria, Ana Piedra-Buena Diaz, Federico Lopez-Moya, Miguel Valverde-Urrea, José Jacobo Zubcoff, José Emilio Martinez-Perez, Javier Lopez-Cepero and Luis Vicente Lopez-Llorca, refers to the banana weevil Cosmopolites sordidus, one of the most serious pests of banana, that causes huge economic losses to worldwide banana production.
The authors conducted 7 field experiments with pitfall traps in commercial orchards naturally infefested with C. sordidus, to develop push-pull control strategies against this pest. The paper provides some interesting information on the use of Volatile organic compounds (VOCs), previously isolated from entomopathogenic and nematophagous fungi, combined with the use of sordidin (agrregation pheromone). The effect of several climatic and production factors (site and season) that can influence the trapping efficacy are also presented. However, in my oppinion, the methodology used do not respond clearly to the hypotheses formulated in the manuscript, regarding the repellent effect of VOCs C1, C2, C5 and C7 (see M Methods), and the results and conclusions cannot thus be fully validated. Additionally, in different Chapters in this paper (Introduction – Line 117 ; Material and methods, Line 123; Discussion, Line 296, 325, 331, ) the authors refer the paper by “Lozano-Soria et al., 2020” published in "Insects" in 2020. However, a “Correction” of this same article was recently re-published (march 2024) in the same jornal: Insects 2024, 15(3), 175; https://www.mdpi.com/2075-4450/15/3/175 Correction: Lozano-Soria et al. Volatile Organic Compounds from Entomopathogenic and Nematophagous Fungi, Repel Banana Black Weevil (Cosmopolites sordidus). Insects 2020, 11, 509. It is therefore considered that the presente paper needs to be updated according to knowledge published in 2024.

Experimental design

The major limitation of this paper is the experimental design, as important information about the experimental design in the seven field experiences is not presented, and the number of repetitions of each treatment is not presented (Line 136-143. The different experiments (1-7) were not carried out under the same conditions, and are therefore the conclusions presented are not comparable.
Four treatments were used in the traps (Line 138-142), however, it is not understood why different VOCs (C1, C2, C5 or C7) were used in the 7 different experiments.
A detailed explanation should be given for the VOCs replacement in the different experiments (50% and 100%), and also why in the third, fourth, or sixth week (Table 1).
A clarification of the statistical analysis carried out, namely on the interaction of the climate effects and the effect on VOCs on insect behavior and capture, should be detailed and clarified.

Validity of the findings

The methodology used do not respond clearly to the hypotheses formulated in the manuscript, regarding the repellent effect of VOCs C1, C2, C5 and C7 , and the results and conclusions cannot thus be fully validated. Additionally, in different Chapters in this paper (Introduction – Line 117 ; Material and methods, Line 123; Discussion, Line 296, 325, 331, ) the authors refer the paper by “Lozano-Soria et al., 2020” published in "Insects" in 2020. However, a “Correction” of this same article was recently re-published (march 2024) in the same jornal: Insects 2024, 15(3), 175; https://www.mdpi.com/2075-4450/15/3/175 Correction: Lozano-Soria et al. Volatile Organic Compounds from Entomopathogenic and Nematophagous Fungi, Repel Banana Black Weevil (Cosmopolites sordidus). Insects 2020, 11, 509. It is therefore considered that the presente paper needs to be updated according to knowledge published in 2024.

Additional comments

Introduction
The Introduction is generally appropriate, with appropriate bibliographical references, but it is not properly structured, as the different subjects (pest, VOCs, etc), come up in a somewhat disorganized way. The scientific writing of Fusarium oxysporum should be corrected: the abbreviation 'Foc' is not a classifier (line 65), and should be placed elsewhere. The objectives of the work should be presented in more detail
According to the actual available bibliography (eg. Bazake et al., 2022 https://www.ejfood.org/index.php/ejfood/article/view/469 the information “Few approaches can be used to manage this pest in an environmentally safe manner” (Line -81-82 ) does not seem to me to be the most appropriate, and should be reviewed.

Material and Methods
-It would be important to know the origin, the host and the year of isolation (Strain number/Culture collection or other) of the isolates used for VOCs used in the work.
-Line 121-123- The reference should be updated to 2024
-Line 124-130-it would be important to clarify the use of different commercial pheromone: Ecosordidina30 and Cosmolure in the different experiences.
-Line 136-148: The area of each banana plantation (Experience 1-7) should be presented; The planting density in each commercial plantation/experience should also be presented; The banana cultivar in each Experience should also be presented. The number of traps per hectar should also be presented (Table 1).
-Empty traps were only used in experiment 7? …. It seems to me that it will be a typo.! (table 1).
-The designation of treatments for VOCs in Table1 (eg. C5 ; VOC C5) should be strandardized
- Line 149-164: Description is a bit confusing. Comprehension would be facilitated if a bibliographic reference on this subject were included.
-In the presentation and discussion of the results, the number of insects captured in the different traps/treatments is never mentioned. This information is very important !

Results and Discussion
For proper understanding, the presentation of the results and the discussion should be reorganized, and new bibliographic references should be included.
The number of insects captured in the different treatments-experiments is not described in the text, nor is it compared with other similar studies, which limits interpretation.
-The Figures and Tables must be self-explanatory, so the Legends can be improved, including information that allows their correct analysis and interpretation (e.g. Figure 2: the meaning of the letters "a" and "b" in the charts should be presented.

·

Basic reporting

This manuscript assessed the efficacy of volatile organic compounds from entomopathogenic and nematophagous fungi against banana weevil under field conditions. The study was conducted in seven field experiments in the Canary Islands between April 2019 and December 2021. I commend the authors for the quality of the data they collected as well as the statistical analysis they performed. However, there is room for improvement in the way authors reported the results. Sometimes, there is no logical flow in some parts of the results section. Also, authors should improve the choice of words as well as their spellings.

Specific comments
Authors consistently used the terminology „mild loci“ across the results section and it is difficult to relate the effect of Pheromone+C1-C7 baited trap on the distribution of the banana weevil.

L53-54: Findings from this study showed that not all VOCs from the biocontrol fungi reduced BW attraction to ist aggregation pheromone. Therefore, authors should only specify VOCs that showed significant repellence to BW. Additionally, authors should mention which of the VOCs they consider as a promising VOC for the management of the pest.
L84-85: The statement should read: …..“when large number of banana plants are infested“
L85: Remove the second word „only“ from the brackets
L107-108: Authors should remove the following sentence „Insect antennae have chemoreceptors capable of detecting VOCs“. Because, insects use in addition to antennae other receptors on their legs for example to detect chemical signals.
L137: How old was each banana field? What was the density of banana plants in each field?
L205: Authors should write „Fungal volatile organic compounds‘ performance varies in their repellency to banana weevil under field conditions“
L208-209: Did authors previously perform experiment about the effect of VOCs on the efficacy of the pheromone or either way?
L217-218: Change „… captures in Experiments 2 and 3 respect to those“ with „…captures in Experiments 2 and 3 compared to those“
L226-227: Replace „only form 3 weeks onward“ with „only from the third week onward“.
L229-230: Authors should change „respect“ with „compared“. Here, the VOC C5. Also, authors should provide a trend regarding the number of insects trapped in the two treatments. Authors should provide legend for Figure 3B. In Figure 3 caption: authors should change „pheromone only traps“ with „pheromone traps only“.
L230-231: Authors said: „No significant differences were found in treatment:time interaction“, which of the results are they referring to?
L232-234. Authors should write: „Pheromone traps only significantly attracted more BWs than traps containing both active compounds (GLMM Negative Binomial, Estimate=1.0046, z-value=3.005, p- value=0.00225; Table S3, Diagram S1 and S2)“.
L238-239: Authors should write: „The number of BWs was significantly lower in traps with fungal VOC C5 only than in traps with pheromone only (GLMM Negative Binomial, Estimate=1.917, z- value=5.036, p-value<0.001; Figure 4, Table S3, Diagram S1 and S2)“.
L242-244: Authors said: „A reduction in captures per trap was found upon the replacement of VOCs in the traps (GLMM Negative Binomial Estimate 0.71488; z-value=1.691, p-value=0.0908). They should rather say: „there was no significant difference in the number of BWs caught across the treatments (GLMM Negative Binomial Estimate 0.71488; z-value=1.691, p-value=0.0908).
L249: „treatment:time“
L251:“and no significant differences were found for treatment…“
L254: „No differences“;
L254: „treatment“ not „treatmen“. Authors should pay attention to the spelling of words throughout the manuscript.
L261-263: Authors should indicate what they man by BW foci or outbreak and how this can be identified on the map? How do authors know that these outbreaks are associated with pheromone baited traps and C5+pheromone traps?
Figure 5 caption: Authors should change the word „point“ to „dot“. For example, yellow points should read „yellow dots“. This should be applied to other Figure captions in the supplementary files (S16-S20).
L265: Authors said „At the end of the experiment, two new mild foci appear in pheromone only containing traps (Figure 5B)“. What does the word „mild foci“ mean? Also, how can someone identify the presence of mild foci in pheromone only containing traps as illustrated on the map? Finally, do authors have a better way of reporting these results. The following sentence is difficult to understand „two mild foci appear in pheromone only containing traps“.
L273: remove „to“ between „associated to with“
L274: Change „next weeks“ with „subsequent weeks“
L272-288: In this paragraph, authors have a reporting style very confusing. For example, they mentioned Figure S16A and then Figure S19 (L275). In L276, authors mentioned results related to Figure S16B. This reporting style is seen throughout the results section and should be significantly improved. My suggestion is that authors should report the results by following the sequence of the number of Tables and Figures. This will enable readers to easily reach your conclusions.
L293: What do you mean by „abolish“? Authors should use another word.
L315-316: Are the authors talking about the strain of B. bassiana itself or the volatile emitted by the fungus? Change „repellent on“ with „repellent to“.
L327-328: Authors said: „In all our field experiments with volatiles BW foci ended closed to pheromone baited traps or completely disappeared“. This is one of the key findings of this study. However, the message is not clear here even in the results section. See comments above.
Additionally, authors said that: „This supports the view that fungal VOCs tested push BW pooling them in the sordidin baited traps“. Authors reported in the results section that not all the VOCs repelled BW. Therefore, the specific VOCs that act as repellent should be mentioned here. Additionally, it is still difficult to understand how the combination of the VOCs with the baited traps will be considered in a push-pull management strategy. In other words, what would be the spatial distribution of both the VOCs and pheromone baited traps in a banana field which will contribute to the management of the weevils.
L330-334: How can authors link the paragraph to their findings? The paragraph is hanging and the discussion should be improved.
L338-339: What do authors mean by: „In our study traps with C5 only captured virtually no BW“?

Experimental design

No comment

Validity of the findings

Find comments above

---

## Round 0.2 · Major Revisions

Dear authors,

As per the report from Reviewer 2 you didn't address all the issues they raised. I urge you, therefore, to revisit the comments made previously and completely address the issues raised before resubmitting.

Reviewer 1 ·

Basic reporting

The manuscript “Volatile organic compounds from entomopathogenic and nematophagous fungi repel banana weevil (Cosmopolites sordidus) under banana field conditions” by Ana Lozano-Soria, Ana Piedra-Buena Diaz, Federico Lopez-Moya, Miguel Valverde-Urrea, José Jacobo Zubcoff, José Emilio Martinez-Perez, Javier Lopez-Cepero and Luis Vicente Lopez-Llorca, refers to the banana weevil (BW) Cosmopolites sordidus, one of the most serious pests of banana, that causes huge economic losses to worldwide banana production. The authors conducted seven field experiments, with pitfall traps in commercial orchards naturally infefested with C. sordidus, to develop push-pull control strategies against BW. The paper provides interesting and importante information on the use of Volatile organic compounds (VOCs), previoussly isolated from entomopathogenic and nematophagous fungi, combined with the use of sordidin (agregation pheromone). The results obtained under banana fiel condition, show that BW repellents have potential to be used in "push-pull" strategies to manage BW sustainably in banana crops.
The authors responded appropriately to all comments and suggestions on how to improve the manuscript.

Experimental design

The authors responded appropriately to all comments and suggestions on how to improve the manuscript.

Validity of the findings

The authors responded appropriately to all comments and suggestions on how to improve the manuscript.

Additional comments

The authors responded appropriately to all comments and suggestions on how to improve the manuscript.

·

Basic reporting

Many thanks for the revised version of the manuscript. However, authors didn't address all the comments raised and therefore this paper cannot be accepted as it is for publication.

Experimental design

n/a

Validity of the findings

n/a

---

## Round 0.3 · Major Revisions

Dear authors,

Before recommending publication or not, I would like to see some aspects of the statistical approach clarified. In your methodology, you state that data did not meet the prerequisite requirements to proceed with an ANOVA and that you have used a non-parametric approach to test captures (Kruskal-Wallis). However, in your results and the graphics provided, you report the ANOVA results and not the Kruskall-Wallis. Also please clarify what post-hoc test you have used after the Kruskal-Wallis test. Also please check your graphs reporting ANOVA results.

In addition, in your methodology you refer to the use of Pearson's correlation, however, I can't see these results reported. Also, you state you have checked the normality of the data, but no results are reported. As you tested before the normal distribution of the residuals for the ANOVA, and these are not normally distributed, the same may happen with the data, and in that case, the non-parametric correlation test should be applied (Spearman's test).

·

Basic reporting

All the comments have been addressed.

Experimental design

All the comments have been addressed.

Validity of the findings

All the comments have been addressed.

---

## Round 0.4 · accepted · Accept

Dear authors,
Thanks for addressing the issues raised. I can now recommend the publication of this manuscript.
Best regards,

Fernando